# Optimal Prediction of the Number of Unseen Species with Multiplicity

**Yi Hao** [§] , **Ping Li**
Cognitive Computing Lab
Baidu Research
10900 NE 8th St. Bellevue, WA 98004, USA
yih179@eng.ucsd.edu , pingli98@gmail.com

## Abstract

Based on a sample of size $n$, we consider estimating the number of symbols that appear at least $\mu$ times in an independent sample of size $a \cdot n$, where $a$ is a given parameter. This formulation includes, as a special case, the well-known problem of inferring the number of unseen species introduced by [Fisher et al.] in 1943 and considered by many others. Of considerable interest in this line of works is the largest $a$ for which the quantity can be accurately predicted. We completely resolve this problem by determining the limit of estimation to be $a \approx (\log n)/\mu$, with both lower and upper bounds matching up to constant factors. For the particular case of $\mu = 1$, this implies the recent result by [Orlitsky et al.] on the unseen species problem. Experimental evaluations show that the proposed estimator performs exceptionally well in practice. Furthermore, the estimator is a linear combination of symbols' empirical counts, and hence linear-time computable.

## 1   Introduction

Let $\Delta$ denote the collection of all discrete distributions. Given an independent sample $X^n$ from an unknown distribution $p \in \Delta$, we are interested in estimating the number of *unseen* symbols that appear *at least $\mu$ times* in an independent sample $Y^m$ from the same distribution $p$, where $m$ is a given sampling parameter. To be more specific, we study the estimation of

$$U_\mu \triangleq U_\mu(X^n, Y^m) \triangleq \sum_x \mathbb{1}_{N_x=0} \cdot \mathbb{1}_{M_x \geq \mu},$$

where the summation is over all symbols $x$ in the potentially unknown alphabet, and $N_x$ and $M_x$ represent the number of times symbol $x$ appearing in $X^n$ and $Y^m$, respectively. In addition, let

$$\Phi_i \triangleq \Phi_i(X^n) \triangleq \sum_x \mathbb{1}_{N_x=i}$$

denote the number of symbols appearing $i$ times in the sample $X^n$. Due to symmetry, the collection of $\Phi_i$'s is a *sufficient statistic* for learning $U_\mu$.

For the special case of $\mu = 1$, the quantity $U_1$ becomes the number of newly observed symbols, and the task reduces to the well-known *unseen species problem*, whose study dates back to more than half a century ago [11]. In the early 1940s, the British chemist and naturalist Corbet had spent two years in Malaya to trap butterflies ($X^n$). Every time he saw a new species, he recorded how many individuals of that species he had trapped ($N_x$).

---

[§]Yi Hao is a Ph.D. candidate in the Department of Electrical and Computer Engineering at the University of California, San Diego. Yi Hao's work was conducted while he was a summer intern at Baidu Research-Bellevue.

After returning to England, Corbet was curious how many new species he would discover if he went back to Malaya for another 2 years. He constructed a table to show the number of species appearing certain number of times ($\Phi_i$), and then presented the problem to Fisher, Father of modern statistics, who provided a remarkable estimator [11] that was later improved on by Good and Toulmin [13],

$$\hat{U}_{\mathrm{GT}} \triangleq -\sum_{i=1}^{n}(-1)^i\Phi_i.$$

Before continuing the discussion, we introduce a convenient notation, $a \triangleq \frac{m}{n}$, and refer to it as the *amplification ratio (or extrapolation ratio)*, a quantity that describes the limit of our estimation power and frequently appears in our subsequent derivations. Henceforth we often write $m$ as $na$ to reflect our intention of inferring future statistics from past observations.

The Good-Toulmin estimator $\hat{U}_{\mathrm{GT}}$ applies to the special case of $a = \mu = 1$ and generalizes to

$$\hat{U}_{\mathrm{GT}} \triangleq -\sum_{i=1}^{n}(-a)^i\Phi_i$$

for general $a$ values, where we kept the same notation. The paper [13] that originally introduced $\hat{U}_{\mathrm{GT}}$ showed that as long as $a \leq 1$, the estimator produces a nearly unbiased estimate with an expected (absolute) deviation of $\mathcal{O}(a\sqrt{n})$, which is negligible as $U_1$ can potentially be $an$.

However, the success of this estimator does not extend to the regime where $a > 1$, both in theory and in practice. For example, an experiment in a recent work [28] addressing $U_1$ estimation shows that even for a simple (shifted) Zipf-law with $p_i \propto 1/(i + 10)$ for $1 \leq i \leq 10^4$, and $a \in [1.3, 1.5]$, estimate $\hat{U}_{\mathrm{GT}}$ can significantly deviate from the actual value and sometimes even becomes negative.

To address this issue, Good and Toulmin applied a smoothing technique to the alternating coefficient sequence $(-a)^i, i = 1, \ldots n$, and replaced it by $(-a)^i \cdot \Pr(\mathrm{bin}(k, 1/(a+1)) \geq i)$ for some properly chosen parameter $k$. The resulting estimator, which [28] terms as the *smoothed Good-Toulmin (SGT)*, has found numerous applications over the past half-century. However, the statistical properties and optimality of this estimator were not well-understood until the recent work of Orlitsky et al. [28, 30].

Noting that the heuristic multiplies the coefficients by the binomial tail probabilities, Orlitsky et al. first proposed the following general smoothing regime. Let $L$ be an independent nonnegative random variable, and denote $\hat{U}_L \triangleq -\sum_{i=1}^{n}(-a)^i \Pr(L \geq i)\Phi_i$. Subsequently, they showed that for $L$ being either a (proper) binomial or Poisson random variable, the induced estimator performs optimally.

Formally and more generally, $U_\mu$ varies from 0 to $m_\mu \triangleq m/\mu = na/\mu$, hence we measure the performance of any $n$-sample estimator $\hat{U}$ by the *worst-case normalized mean-square error* (NMSE),

$$\mathcal{E}_{n,a}^{\mu}(\hat{U}) \triangleq \max_{p \in \Delta} \mathbb{E}_p \left(\frac{\hat{U} - U_\mu}{m_\mu}\right)^2.$$

The main result of [28] states that for some Poisson or binomially distributed $L$, the estimator satisfies $\mathcal{E}_{n,a}^1(\hat{U}_L) \lesssim 1/n^{1/a}$, where $\alpha \lesssim \beta$ abbreviates $\alpha = O(\beta)$. On the other hand, for some absolute constants $c, c' > 0$, and any $a \geq c$ and estimator $\hat{U}$, one has $\mathcal{E}_{n,a}^1(\hat{U}_L) \gtrsim 1/n^{c'/a}$, where $\alpha \gtrsim \beta$ abbreviates $\alpha = \Omega(\beta)$. Combined, the results established the optimality of the smoothing scheme.

## 1.1 New results

As described previously, the problem that we study here is natural generalization of the unseen species problem, and calls for estimating $U_\mu$, the number of symbols that appear at least $\mu$ times in the future sample. One motivation for considering this problem is reproducibility. For example, in the aforementioned butterfly trapping story, a basic task one often wants to perform is checking the existence of *sexual dimorphism* in a newly observed butterfly species, meaning that the two sexes look completely different. This clearly requires inferring $U_\mu$ for $\mu \geq 2$.

Replacing butterflies by words, *vocabulary size estimation* [8, 10, 18, 34] aims to determine how many words a writer, say William Shakespeare, knew based on his written works. An intuitive and widely used approach is to simply add up the number of observed (distinct) words and some estimate

of $U_1$. With the same motivation, we may also want to know how many words fall into a writer's *common vocabulary* (excluding those that appear only once or twice), which calls for estimating $U_\mu$.

For another example, app developers are often interested in knowing how many new users their apps will have in a future time period. In addition, they usually care more about active users who will use the apps for at least a certain number of times. Under appropriate assumptions, this again translates to a $U_\mu$ estimation problem. The same rationale applies to many other types of businesses such as advertising, catering, and entertainment, since $U_\mu$ is a natural business growth indicator.

There has been a long line of research works on estimating $U_1$, in which of considerable interest is the largest $a$ for which the quantity can be accurately predicted. The generalization of this "unseen species" problem, on the other hand, is a new problem that we propose and rigorously study. As the subsequent discussion shows, we completely resolved this problem by determining the limit of estimation to be $a \approx (\log n)/\mu$, with both lower and upper bounds matching up to constant factors.

Our estimator is linear-time computable given $\Phi_i$'s, and has the form

$$\hat{U}_\mu \triangleq \hat{U}_\mu(X^n, a) \triangleq \sum_{i=1}^n s_i \cdot \Phi_i,$$

where for $r \sim \frac{\log n}{a}$, the *smoothing rate*, we denote the *$i$-th smoothed coefficient* by

$$s_i \triangleq - \sum_{j=0}^{(\mu-1)\wedge i} (-a)^i (-1)^j \binom{i}{j} \Pr(\text{Poi}(r) \geq i+j),$$

where the notation $\alpha \wedge \beta$ abbreviates $\min\{\alpha, \beta\}$. In particular, if we set $\mu = 1$, the estimator reduces to $\hat{U}_L$ in [28] with $L \sim \text{Poi}(r)$. Interestingly, if we remove the smoothing probability, the estimator becomes the vanilla Good-Toulmin estimator $\hat{U}_{\text{GT}}$ when $\mu = 1$, and becomes the following "generalized Good-Toulmin" estimator (denoted by $\hat{U}_{\text{GGT}}$) when $\mu \geq 1$:

$$\hat{U}_{\text{GGT}} \triangleq (-1)^\mu \sum_{i=\mu}^n (-a)^i \binom{i-1}{\mu-1} \Phi_i.$$

The deviation of $\hat{U}_{\text{GGT}}$ can be found in Appendix J.

The following two results essentially establish the optimality of $\hat{U}_\mu$ and determine the min-max learning risk of approximating $U_\mu$. In particular, these theorems imply the main result of [28] as a special case. The first theorem bounds the worst-case NMSE of our estimator.

**Theorem 1.** *There exist absolute constants $c$ and $c_0$ such that for any parameter $a \in [1, (c \log n)/\mu]$, the estimator $\hat{U}_\mu$ described above satisfies*

$$\mathcal{E}_{n,a}^\mu(\hat{U}_\mu) \lesssim \frac{1}{n^{c_0/a}}.$$

The second theorem lowerly bounds the worst-case NMSE of the best estimator.

**Theorem 2.** *There exist absolute constants $c'$ and $c_0'$ such that for any $a \geq \max\{1, (c' \log n)/\mu\}$, and any $n$-sample estimator $\hat{U}$,*

$$\mathcal{E}_{n,a}^\mu(\hat{U}) \gtrsim \frac{1}{n^{c_0'/a}}.$$

## 1.2  Related Work

**Unseen species problem**   Initiated by the seminal work of [11, 13], species estimation is an important problem in a variety of scientific disciplines. Over the past few decades, the problem has found numerous applications to the estimation of ecological or bacterial diversity [3, 4, 6, 9], bacterial and microbial diversity [12, 27, 29], and writers' vocabulary sizes [10, 34, 8], and the study of database attribute variations [14], immune receptors [31], and genetic variations [23].

A number of estimators were proposed, notable ones include the Good-Toulmin (GT) estimator [13, 28, 30], jackknife estimator [32], abundance coverage estimator (ACE) [5], Chao-Lee estimator [4, 6], and a linear-programming based (LP) estimator [35]. Even just for this $U_1$ estimation task, the GT estimator and their variants are the only ones known [28, 30] to achieve the optimal worst-case NMSE. The ACE and Chao-Lee estimators are designed for uniform distributions, and LP estimator has exponentially worse NMSE guaranty and runs in polynomial time, inefficient compared with GT or others in [28]. Another way of generalizing this unseen species problem is to consider the setting involving multiple species groups (hence multiple distributions). Paper [16] recently studied this problem as an application of its proposed multi-distribution functional estimation methodology.

**Functional estimation** In recent years, a related area that has seen significant advances is distribution functional estimation. For example, one may want to infer the entropy of an unknown distribution from its i.i.d. sample. Other important functionals include support size, $L_1$ distance to a fixed distribution, general Lipschtiz functionals, and *support coverage*, the expected number of distinct symbols in a new sample. Two types of methods have been proposed: one first finds an approximation of the targeted functional, then find a near-unbiased estimator for the proxy [19, 20, 22, 25, 28, 38, 39]; the other focuses on symmetric functionals, and tackles the problem by computing an estimate of the distribution probability multiset and plugging it into the functional [2, 7, 15, 17, 21, 26, 35, 36].

Our approach essentially falls into the first category, and is closer to [16, 22, 28] utilizing smoothing methods based on Bessel functions, with which we compare carefully in Section 3. Other methods either require assumptions not satisfied in our setting, such as Lipschitzness [20, 17] or the existence of highly concentrated estimators [2, 17]; or not known to work (well) even for $U_1$ estimation, such as those based on minimax polynomials [25, 26, 38, 39] or linear programming [15, 35, 36] (see above).

**Outline** For the rest of the paper, we mainly focus on constructing the estimator and deriving the upper bound. The proofs of the lower bound and technical lemmas are postponed to the supplementary. In Section 2, we present a roadmap for our construction and analysis, and show that Poissonization works well in simplifying the problem. In Section 3, we utilize connections between Bessel functions and Poisson probabilities to approximate the expectation of $U_\mu$, and introduce $r$ as a tunable hyper-parameter controlling the variance-bias tradeoff. Section 4 then finds the explicit form of the estimator and outlines the tools and key lemmas for the bias analysis. Finally, we numerically evaluate the performance of the estimator in Section 5, demonstrating the excellent practicality of our approach.

## 2   MSE Decomposition and Poisson Sampling

First we address the upper bound in Theorem 1, decompose the problem into several parts, and simplify our reasoning with *Poisson sampling*, a technique to remove symbol-count dependence. Given $n$, $a$, and $\mu$, the normalizing factor $M_\mu = na/\mu$ is simply a constant. Hence, it suffices to bound the mean squared error (MSE) of the estimator. On a high level, our analysis follows by

$$\text{MSE} \triangleq \mathbb{E}\left(\hat{U}_\mu - U_\mu\right)^2 \leq 2\text{Var}(\hat{U}_\mu) + 2\text{Var}\left(U_\mu\right) + \left(\mathbb{E}[\hat{U}_\mu] - \mathbb{E}\left[U_\mu\right]\right)^2,$$

based on which we construct $\hat{U}_\mu$ and bound each term in the last line separately.

**Variances** As one might expect, a difficulty in analyzing the variance is caused by dependence. Given a sample $X^n$ from some unknown distribution $p$, our estimator takes the form of $\sum_i s_i \Phi_i$ where $\Phi_i$ denotes the number of symbols appearing $i$ times in the sample. While $X_i$'s are independent by our assumption, $\Phi_i$'s are dependent random variables, and putting involved coefficients in front of them certainly makes the analysis even harder. A similar difficulty appears when we consider the target random variable $U_\mu$. To get around these obstacles, we apply Steele's inequality [33].

**Lemma 1** (Steele's inequality). *If $S(x^n)$ is any real function of $n$ variables and if $X_i, X_i', 1 \leq i \leq n$ are $2n$ i.i.d. random variables then $\text{Var}\left(S\right) \leq \frac{1}{2}\mathbb{E}\left[\sum_{i=1}^n (S - S_i)^2\right]$, where $S \triangleq S(X^n)$ and $S_i$ is given by replacing the $i$-th observation with $X_i'$.*

The inequality basically states that as long as we have i.i.d. input $X^n$ and care only a functional $S(X^n)$ *that is not too sensitive to input changes*, the variance of $S(X^n)$ will be reasonably bounded. For example, $U_\mu$ is a functional of the length-$(n+m)$ i.i.d. sample $(X^n, Y^m)$, where modifying a

single symbol changes $U_\mu$ by at most 1. Hence, Lemma 1 bounds its variance as

$$\mathrm{Var}(U_\mu) \leq \frac{1}{2}(n+m).$$

In the supplementary material, we derive a similar bound of $\mathrm{Var}(\hat{U}_\mu) \leq 2n \max_i s_i^2$ together with $|s_i| \leq e^{(2a-1)r}$ under appropriate assumptions, which requires significantly more work.

**Absolute bias**    Analyzing the bias term $\mathbb{E}[\hat{U}_\mu - U_\mu]$ directly relates to the estimator construction. Intuitively, we first find the exact form of $\mathbb{E}[U_\mu]$, which induces to an unbiased $n$-sample estimator. Then, we gradually increase the absolute bias, with an intention to reduce the variance (or the coefficient differences by the previous reasoning). The smoothing rate $r$ appearing in the coefficient expressions is a tunable hyper-parameter for this variance-bias tradeoff. In fact, this is the only hyper-parameter for our algorithm, which implies an implementation advantage. See Section 3 for details on how we approximate the target quantity and how $r$ comes into the picture, and Section 4 for how our approximation induces estimator $\hat{U}_\mu$ and how we bound its bias.

To simplify our bias analysis, we also consider a closely related statistical model where the corresponding sample sizes are respectively $N \sim \mathrm{Poi}(n)$ and $N' \sim \mathrm{Poi}(m)$, independent of the samples. This leads to a technique in statistical learning, commonly known as *Poisson sampling*. The reason for employing this *Poissonization* procedure is that it eliminates the dependence between sample counts and turns a binomial probability $\binom{b}{a} q^a (1-q)^{b-a}$ to the simpler Poisson version $e^{-bq}(bq)^a/a!$.

Henceforth, we indicate Poissonization by attaching a letter ᴘ to an expression, for example, $U_\mu^{\mathrm{P}}$ for $U_\mu$, and $\hat{U}_\mu^{\mathrm{P}}$ for $\hat{U}_\mu$, under the Poisson model. The following lemma partially justifies our reasoning.

**Lemma 2.** *For any $n$, $a$ and $p$, we have* $\left|\mathbb{E}[U_\mu - U_\mu^{\mathrm{P}}]\right| \leq 4$.

A similar result holds for the estimator part.

**Lemma 3.** *Under the conditions in Theorem 1, we have* $|\mathbb{E}[\hat{U}_\mu - \hat{U}_\mu^{\mathrm{P}}]| \leq 2e^{(2a-1)r}$.

As we show latter in the paper, this bound is also tiny compared with the normalizing factor $m_\mu$.

## 3   Bessel Functions and Approximation of Unseen

Following the reasoning in the last section, Poissonizing the samples facilitates our derivation without affecting our results. In this section, we will assume that the sample sizes are $\mathrm{Poi}(n)$ and $\mathrm{Poi}(m)$ for the past and future samples. To not further complicate the notation, we still use $N_x$'s and $M_x$'s to denote the counts of symbols in the samples. An important point to note is that now they are all *mutually independent* Poisson random variables.

We introduce a new notation $\lambda_x \triangleq np_x$ for every symbol $x$. Now, we can write $N_x \sim \mathrm{Poi}(\lambda_x)$ and $M_x \sim \mathrm{Poi}(a\lambda_x)$. For the target quantity $U_\mu^{\mathrm{P}}$, taking expectation yields

$$\mathbb{E}\left[U_\mu^{\mathrm{P}}\right] = \sum_x \mathbb{E}\left[\mathbb{1}_{M_x \geq \mu}\right] \cdot \mathbb{E}\left[\mathbb{1}_{N_x = 0}\right]$$

$$= \sum_x \left(1 - e^{-a\lambda_x} \sum_{j \leq \mu-1} \frac{(a\lambda_x)^j}{j!}\right) \cdot e^{-\lambda_x}$$

$$\triangleq \sum_x F_\mu(a\lambda_x) \cdot e^{-\lambda_x},$$

where the definition of function $F_\mu$ is clear from the context. Leveraging Lemma 1 and the standard exponential tail bounds on Poisson variables, we can show that $U_\mu^{\mathrm{P}}$ has small variance. In other words, it suffices to approximate the above expectation. Next, express function $F_\mu(y)$ into the summation of $\mu$ terms, with each being a polynomial of $e^{-y}$ and $y$ that vanishes at $y = 0$,

$$F_\mu(y) = (1 - e^{-y}) + \sum_{1 \leq j \leq \mu-1} \left(-\frac{e^{-y}y^j}{j!}\right).$$

**Bessel functions**  We approximate terms in the parentheses by polynomial series induced by the Bessel functions. For completeness, we briefly introduce this function class. The *Bessel functions of the first kind* $J_s(y)$ are defined as the solutions to the Bessel differential equation

$$y^2 \frac{d^2 z}{dy^2} + y \frac{dz}{dy} + (y^2 - s^2)z = 0,$$

for $s \in \mathbb{Z}^+$, which is referred to as the *order* of the function.

For every $s$, function $J_s(y)$ admits a series expansion [24] at $y = 0$,

$$J_s(y) = \sum_{t=0}^{\infty} \frac{(-1)^t}{t!(t+s)!} \left(\frac{y}{2}\right)^{2t+s}.$$

For our purpose, it is sufficient to use only the even order Bessel functions. For $j \geq 0$, denote

$$g_j(y) \triangleq J_{2j}(2\sqrt{y}) = \sum_{t=0}^{\infty} \frac{(-1)^t y^{t+j}}{t!(t+2j)!}.$$

As we mentioned in Section 1, the problem reduces to the unseen species problem for $\mu = 1$. Hence, we follow the smoothing technique in [28] and approximate the first term $(1 - e^{-y})$ by

$$f_0(y) \triangleq (1 - e^{-y}) - e^{-y} \left[ e^{-r} \int_0^y g_0(sr)e^s ds \right] = -\sum_{j=1}^{\infty} \frac{(-y)^j}{j!} \Pr(\mathrm{Poi}(r) \geq j),$$

where $r$ is the aforementioned tunable hyper-parameter to balance the variance and bias. For the $j$-th term in the expression of $F_\mu$, we follow the technique in [22] and approximate every $-e^{-y}y^j$ by

$$f_j(y) \triangleq -e^{-y}y^j + \int_r^{\infty} e^{-\alpha} \alpha^j g_j(\alpha y)d\alpha = -\int_0^r e^{-\alpha} \alpha^j g_j(\alpha y)d\alpha,$$

where the second equality follows by the lemma below.

**Lemma 4.**  *[22] For any $j \in \mathbb{Z}^+$ and $y \geq 0$,*

$$e^{-y}y^j = \int_0^{\infty} e^{-\alpha} \alpha^j g_j(\alpha y)d\alpha.$$

In other words, function $f_i(y)$ corresponds to the truncated integral form of $e^{-y}y^j$ at level $r$, where the integral is expressed in terms of the Bessel functions.

It is worth mentioning that the result in [22] (induced by the use of Bessel functions) and that in our paper are orthogonal to a certain extend. The result in [22] addresses the problem of estimating a distribution functional that is roughly Lipschitz, and designs an $n$-sample estimator that performs as well as the $n\sqrt{\log n}$-sample MLE estimator. The problem we consider here is estimating a random distribution functional that depends on the current sample. Even if we consider the functional's expectation and divide it by our normalization factor $m_\mu$, the transformed functional is not Lipschitz. In addition, our extrapolation factor $a$ can be as large as $(\log n)/\mu$, which, for constant $\mu$ values, is of order $\log n$. Paper [16] generalizes the results of [22] to functionals involving multiple distributions, such as the total variation distance, but does not strengthen those results for the one-distribution case.

Another paper that studies the same problem as [22] is [20]. In terms of techniques, [20] leverages integral forms of the best approximation polynomials, and does not rely on smoothing techniques involving Bessel functions. In addition, [20] applies its smoothing technique to only probabilities smaller than $(\log n)/n$, while we apply the aforementioned technique to the entire range $[0, 1]$. In terms of results, [20] requires the functional to be Lipschitz, which is not the case for our problem, and improves the $\sqrt{n}$ factor in [22] to $\log n$, which is not the right factor for our problem. The paper also addresses the problem of estimating $\mathbb{E}[U_1]$, while the estimator is essentially the same as [28]. As the above derivation shows, the corresponding technique only approximates $(1 - e^{-y})$, and by the reasoning in [22], does not address other $e^{-y}y^j$ terms.

**Approximation function**  Consequently, we will approximate $F_\mu(y)$ by

$$\tilde{F}_\mu(y) \triangleq \sum_{j \leq \mu - 1} \frac{f_j(y)}{j!},$$

which implicitly depends on $r$ and approaches $F_\mu(y)$ as $r \to \infty$. Our estimator simply estimates a variant of this function nearly unbiasedly. Intuitively, if $r$ is large, the bias is small, e.g., for $r = \infty$, the bias is zero, but the variance, as shown by the example in Section 1, can be fairly large. On the other hand, if $r$ is small, e.g., for $r = 0$, the variance is brought down to zero but the bias is large.

## 4 Estimator Construction and Bias Analysis

**Approximation and estimator**  Following the previous section, we approximate the value of $U_\mu^{\mathrm{P}}$ by a *near-unbiased* estimator of

$$\tilde{U}_\mu^{\mathrm{P}} \triangleq \sum_x \tilde{F}_\mu(a\lambda_x) \cdot e^{-\lambda_x}.$$

The statistics available to us are essentially indicator functions $\mathbb{1}_{N_x=t}$, or $\frac{\lambda_x^t}{t!} \cdot e^{-\lambda_x}$ in expectation. This suggests we expand each term in the summation as the product of $e^{-\lambda_x}$ and a polynomial series. Equivalently, we find the series expansion of $\tilde{F}_\mu(y)$.

**Lemma 5.** *For any non-negative number $y$,*

$$\tilde{F}_\mu(y) = -\sum_{i=1} c_i \cdot \frac{y^i}{i!},$$

*where the coefficients are defined as*

$$c_i \triangleq (-1)^i \Pr(\mathrm{Poi}(r) \geq i) + \sum_{j=1}^{(\mu-1)\wedge i} (-1)^{i-j} \binom{i}{j} \Pr(\mathrm{Poi}(r) \geq i + j + 1).$$

We postpone the proof of Lemma 5 to the supplementary material, which relies on Fubini's theorem and the series expansions of the incomplete Gamma functions centered at the origin. Next, for the target quantity $\tilde{U}_\mu$, substituting each $\tilde{F}_\mu$ in the expression in Lemma 5 yields

$$\tilde{U}_\mu^{\mathrm{P}} = \sum_x \tilde{F}_\mu(a\lambda_x) \cdot e^{-\lambda_x} = -\sum_x \sum_{i=1}^\infty (c_i a^i) \cdot e^{-\lambda_x} \frac{\lambda_x^i}{i!} = -\sum_{i=1}^\infty (c_i a^i) \cdot \mathbb{E}[\Phi_i].$$

It should be clear that $s_i \approx c_i a^i$ for every $i$ as they are equal if we replace each smoothing probability $\Pr(\mathrm{Poi}(r) \geq i+j+1)$ by $\Pr(\mathrm{Poi}(r) \geq i+j)$ in the expression of $c_i$. Hence, given sample statistics $\Phi_i$ for $i \geq 1$, a *near-unbiased* estimator of $\tilde{U}_\mu^{\mathrm{P}}$ is

$$\hat{U}_\mu^{\mathrm{P}} = -\sum_{i=1}^\infty s_i \cdot \Phi_i.$$

More concretely, the absolute bias satisfies $|\mathbb{E}[\hat{U}_\mu^{\mathrm{P}}] - \tilde{U}_\mu^{\mathrm{P}}| \leq na \cdot \Pr(\mathrm{Poi}(r) \leq \mu)$, which is proved in the supplementary material leveraging point-wise bounds on Bessel functions.

**Bias analysis**  Unifying the previous bounds by the triangle inequality, the absolute bias of the final estimator $\hat{U}_\mu$ admits the decomposition

$$\mathrm{Bias} \triangleq \left| \mathbb{E}[\hat{U}_\mu] - \mathbb{E}[U_\mu] \right| \leq \left| \mathbb{E}[\hat{U}_\mu] - \mathbb{E}[\hat{U}_\mu^{\mathrm{P}}] \right| + \left| \mathbb{E}[\hat{U}_\mu^{\mathrm{P}}] - \tilde{U}_\mu^{\mathrm{P}} \right| + \left| \tilde{U}_\mu^{\mathrm{P}} - \mathbb{E}[U_\mu^{\mathrm{P}}] \right| + \left| \mathbb{E}[U_\mu^{\mathrm{P}}] - \mathbb{E}[U_\mu] \right|.$$

Then, the first term on the right-hand side is at most $2e^{(2a-1)r}$ by Lemma 3, the second is at most $na \cdot \Pr(\mathrm{Poi}(r) \leq \mu)$ as stated in the last subsection, and the last is at most 4 by Lemma 2. As for the third term, our construction of function $\tilde{F}_\mu$ and $f_j$ yields

$$\begin{aligned}
\mathrm{Bias}_3 &\triangleq \left| \tilde{U}_\mu^{\mathrm{P}} - \mathbb{E}[U_\mu^{\mathrm{P}}] \right| \\
&\leq \sum_x |\tilde{F}_\mu(a\lambda_x) - F_\mu(a\lambda_x)| \\
&\leq \sum_x \left( e^{-r} \int_0^{a\lambda_x} |g_0(sr)| e^{s-a\lambda_x} ds + \sum_{j \leq \mu-1} \frac{1}{j!} \int_r^\infty |g_j(\alpha a\lambda_x)| e^{-\alpha} \alpha^j d\alpha \right).
\end{aligned}$$

Analyzing the last summation calls for point-wise and uniform bounds on $|g_j(y)|$, which further requires bounds on the Bessel functions by $g_j(y) = J_{2j}(2\sqrt{y})$. It turns out that Bessel functions, while being transcendental, behave nicely for nonnegative reals.

**Lemma 6.** *[1, 37] For any $s, y \geq 0$, we have $|J_s(y)| \leq 1$ and $|J_s(y)| \leq \frac{(y/2)^s}{s!}$.*

Building on this lemma and under appropriate conditions, the supplementary material shows that

$$\text{Bias}_3 \leq an(\mu + 1) \cdot \Pr(\text{Poi}(r) \leq \mu).$$

The proof of Theorem 1 follows from consolidating these bias bounds and the previous variance bounds by the MSE decomposition at the beginning of Section 2.

## 5 Experiments

Given relevant parameters and sample $X^n \sim p$, our estimator computes a linear combination of $\Phi_i$'s, the number of symbols appearing exactly $i$ times in $X^n$. The estimator is both easy to implement and near-linear-time computable. To demonstrate the effectiveness of our approach, we present in this section experimental results on synthetic data (see the plots on the next page), and in the supplementary material ones on real data.

**Hyper-Parameters**  Our algorithm has a single hyper-parameter $r$. In the experiments, we choose

$$r = \frac{\log(n(a+1)^2/(a-1))}{2a}.$$

This equals the $r$ value in Theorem 1, up to absolute constant factors, and is consistent with [28]. In fact, our theorem holds for both choices of $r$ and the value in Theorem 1 is for proof simplification.

**Distributions**  We consider a support size of $S \triangleq 10{,}000$ and six distributions: a uniform distribution; a two-step distribution with half the symbols having probability $1/(2S)$, and the other half having probability $3/(2S)$; two Zipf distributions with parameter $\alpha \triangleq 0.5$ and 1, satisfying $p_i \propto i^{-\alpha}$, truncated at $i = S$ and renormalized; a distribution generated by the uniform prior over $\Delta_S$, the $S$-dimensional simplex; a distribution generated by a Dirichlet-$1/2$ prior.

**Experimental settings**  We fix the sample size to be $n = S/2$, vary $a$ from 1 to 10, and test three different $\mu$ values, $\mu = 1$, 2, and 3. The (expected) true value $\mathbb{E}[U_\mu]$ is shown in the black dashed line, and our estimator is shown in red and blue, with the solid red line representing its mean estimate, and the shaded blue area corresponding to one standard deviation, both based on 100 independent simulations. As these plots demonstrate, our simple estimator performs well in numerous settings. In addition, for $\mu = 1$, our estimator reduces to that in [28]. In the work of [28], this estimator has been compared with several state-of-the-art estimators and outperformed them in nearly every experiment.

In the supplementary, we conduct experiments on larger $\mu$ values. We observe that our estimator can provide meaningful results for $\mu = 5$ on the tested distributions. However, as $\mu$ further increases to $8 \approx \log n$, the estimator does not produce useful estimates. Both of these are consistent with our theoretical justifications, as the optimal threshold for $a$ is $\Theta((\log n)/\mu)$.

## Broader Impact

In this paper, we propose and completely resolve a natural generalization of the well-known unseen species problem. The algorithm we construct is simple in its form and linear-time computable, and performs fairly well on different synthetic and real datasets. We fully generalize the notable work of [28] in terms of the formulation, algorithm, and main results (MSE bounds). As illustrated in Section 1.1 and 1.2, our algorithm has numerous potential applications, such as active app user estimation, vocabulary size estimation, business growth analysis, species diversity estimation, database attribute study, and genetic variation study. A possible downside is that the formulation ignores any prior information, and only assumes that the distribution is discrete. A promising strategy is to incorporate the Bayesian inference framework, which will be our future research direction.

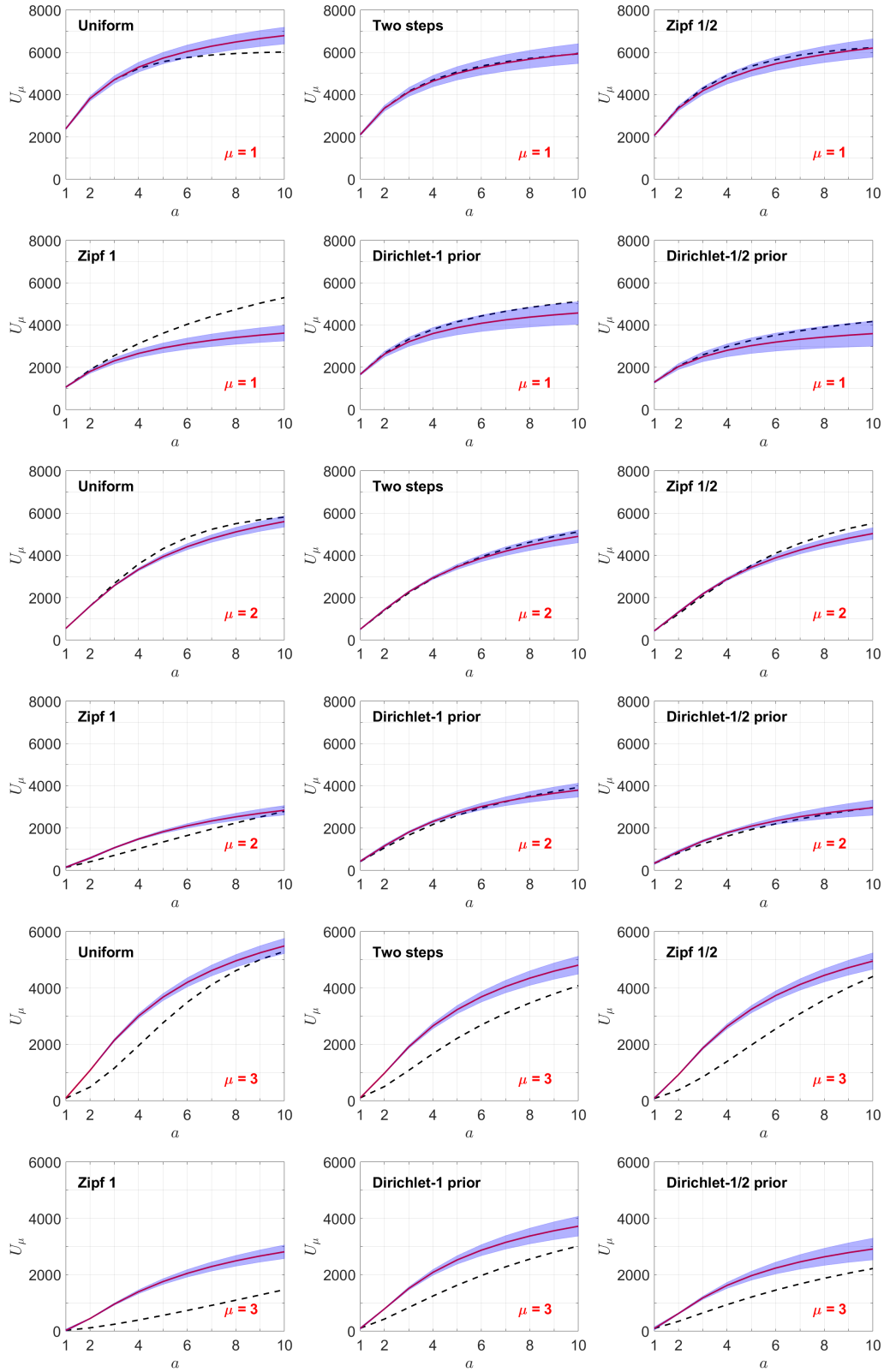

Figure 1: We present three plot sets, for $\mu = 1, 2$, and $3$, each containing 6 plots, for 6 different distributions. The dashed (black) curves represent the truth while the solid (red) curves represent our estimate (the mean). The shaded areas illustrate the standard deviation from the mean.

## Acknowledgments and Disclosure of Funding

We would like to thank the anonymous reviewers and meta-reviewer for their thoughtful and helpful suggestions and comments. Funding in direct support of this work: Baidu Research - Bellevue. Additional revenues related to this work: Not applicable.

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
