[Supplementary Material]

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

As for estimator $\hat{U}_\mu = \sum_{i \geq 1} s_i \Phi_i = \sum_{i \geq 1} s_i (\sum_x \mathbb{1}_{N_x = i})$, where $N_x$ denotes the number of times symbol $x$ appearing in the sample, changing a single symbol changes its value by at most $2 \max_i |s_i|$.

As we will see in Appendix C, the maximum magnitude of the coefficients is bound by $2e^{(2a-1)r}$ from above, implying an upper bound of $8ne^{2(2a-1)r}$ on the variance of our estimator.

# B Proof of Lemma 2

Recall that $U_\mu$ is the quantity of interest, and $U_\mu^{\mathrm{P}}$ is its alternative under Poisson sampling.

**Lemma 2.** *For any $n$, $a$ and $p$, we have $\left|\mathbb{E}[U_\mu - U_\mu^{\mathrm{P}}]\right| \leq 4$.*

*Proof.* The original symbols counts are defined as $N_x$ and $M_x$ in the main paper. For simplicity, we will use $N_x'$ and $M_x'$ to denote their Poissonized versions. Then the absolute difference becomes

$$\left|\mathbb{E}[U_\mu - U_\mu^{\mathrm{P}}]\right| = \left|\sum_x \mathbb{E}[\mathbb{1}_{N_x=0} \cdot \mathbb{1}_{M_x \geq \mu} - \mathbb{1}_{N_x'=0} \cdot \mathbb{1}_{M_x' \geq \mu}]\right|.$$

By symmetry, it suffices to bound a single term in the summation.

$$
\begin{aligned}
D(p_x) &\overset{(a)}{:=} \left|\mathbb{E}\left[\mathbb{1}_{N_x=0} \cdot \mathbb{1}_{M_x \geq \mu} - \mathbb{1}_{N_x'=0} \cdot \mathbb{1}_{M_x' \geq \mu}\right]\right| \\
&\overset{(b)}{=} \left|\mathbb{E}\left[\mathbb{1}_{M_x \geq \mu} \cdot \left(\mathbb{1}_{N_x=0} - \mathbb{1}_{N_x'=0}\right)\right] + \mathbb{E}\left[\left(\mathbb{1}_{M_x \geq \mu} - \mathbb{1}_{M_x' \geq \mu}\right) \cdot \mathbb{1}_{N_x'=0}\right]\right| \\
&\overset{(c)}{=} \left|\mathbb{E}\left[\mathbb{1}_{M_x \geq \mu}\right] \cdot \mathbb{E}\left[\left(\mathbb{1}_{N_x=0} - \mathbb{1}_{N_x'=0}\right)\right] + \mathbb{E}\left[\left(\mathbb{1}_{M_x \geq \mu} - \mathbb{1}_{M_x' \geq \mu}\right)\right] \cdot \mathbb{E}\left[\mathbb{1}_{N_x'=0}\right]\right| \\
&\overset{(d)}{\leq} \left|\mathbb{E}\left[\mathbb{1}_{N_x=0} - \mathbb{1}_{N_x'=0}\right]\right| + \left|\mathbb{E}\left[\mathbb{1}_{M_x \geq \mu} - \mathbb{1}_{M_x' \geq \mu}\right]\right| \\
&\overset{(e)}{\leq} 4p_x,
\end{aligned}
$$

where $(a)$ defines $D(p_x)$; $(b)$ follows by subtracting and adding the term $\mathbb{1}_{N_x'=0} \cdot \mathbb{1}_{M_x \geq \mu}$; $(c)$ follows by independence between the empirical counts; $(d)$ follows by the fact that the expectation of an indicator function is at most 1; $(e)$ follows by the following inequality:

**Lemma 7.** *[3] For any $q \in [0,1]$, $Y \sim \mathrm{Poi}(mq)$, $Z \sim \mathrm{bin}(m,q)$, and real function $f$,*
$$\left|\mathbb{E}\left[f(Y)\right] - \mathbb{E}\left[f(Z)\right]\right| \leq 2q \sup_j |f(j)|.$$

Therefore, the difference we want to bound is at most

$$\left|\mathbb{E}[U_\mu - U_\mu^{\mathrm{P}}]\right| \leq \sum_x D(p_x) \leq \sum_x 4p_x = 4. \qquad \square$$

# C Proof of Lemma 3 and Coefficient Bound

Recall that our estimator takes the form of

$$\hat{U}_\mu \triangleq \hat{U}_\mu(X^n, a) \triangleq \sum_{i=1}^n s_i \cdot \Phi_i.$$

where $\Phi_i = \sum_x \mathbb{1}_{N_x=i}$ denotes the number of symbols appearing exactly $i$ times. Similar to the last section, we will use $N_x'$ and $\Phi_i'$ to denote the Poissonized versions of these quantities.

**Lemma 3.** *Under the conditions in Theorem 1, we have $|\mathbb{E}[\hat{U}_\mu - \hat{U}_\mu^{\mathrm{P}}]| \leq 2e^{(2a-1)r}$.*

*Proof.* Adopting the Poisson sampling model changes the expectation of the estimator by

$$
\begin{aligned}
|\mathbb{E}[\hat{U}_\mu - \hat{U}_\mu^{\mathrm{P}}]| &= \left|\sum_i (s_i \cdot \Phi_i - s_i \cdot \Phi_i')\right| \\
&\overset{(a)}{\leq} \sum_x \left|\sum_i s_i \cdot \left(\mathbb{1}_{N_x=i} - \mathbb{1}_{N_x'=i}\right)\right| \\
&\overset{(b)}{\leq} \sum_x 2p_x \max_i |s_i| \\
&\overset{(c)}{=} 2 \max_i |s_i|,
\end{aligned}
$$

where $(a)$ follows by the triangle inequality; $(b)$ follows by Lemma 7; $(c)$ follows by $\sum_x p_x = 1$.

To complete our proof of Lemma 3, we establish the following bounds on the estimator's coefficients. Intuitively, this captures how sensitive the estimator is to changes in the input sample.

**Lemma 8.** *For any positive index $i \in \mathbb{N}$,*

$$|s_i| \le e^{(2a-1)r}.$$

*Proof of Lemma 8.* For any positive index $i \in \mathbb{N}$,

$$
\begin{aligned}
|s_i| &\overset{(a)}{\le} a^i \sum_{j=0}^{(\mu-1)\wedge i} \frac{i!}{j!(i-j)!} \Pr(\text{Poi}(r) \ge i+j) \\
&\overset{(b)}{\le} a^i \Pr(\text{Poi}(r) \ge i) \left( \sum_{j=0}^{(\mu-1)\wedge i} \frac{i!}{j!(i-j)!} \right) \\
&\overset{(c)}{\le} (2a)^i \Pr(\text{Poi}(r) \ge i) \\
&\overset{(d)}{=} (2a)^i e^{-r} \sum_{j=i}^{\infty} \frac{r^j}{j!} \\
&\overset{(e)}{\le} e^{-r} \sum_{j=i}^{\infty} \frac{(2ar)^j}{j!} \\
&\overset{(f)}{\le} e^{(2a-1)r}.
\end{aligned}
$$

where $(a)$ follows by the definition of $s_i$ and the triangle inequality; $(b)$ follows by the monotonicity of Poisson tail probabilities; $(c)$ follows by the binomial expansion of $(1+1)^i$; $(d)$ follows by the expressions of Poisson probabilities; $(e)$ follows by the monotonicity of Power functions; and $(f)$ follows by the series expansion of the function $\exp(x)$.

Applying this lemma to the right-hand side of our bound on $|\mathbb{E}[\hat{U}_\mu - \hat{U}_\mu^{\text{P}}]|$ completes the proof. $\quad\square$

# D    Proof of Lemma 4

Recall that $J_j$ denotes the $j$-th order Bessel function of the first kind. For $j \ge 0$, we have

$$g_j(y) \triangleq J_{2j}(2\sqrt{y}) = \sum_{t=0}^{\infty} \frac{(-1)^t y^{t+j}}{t!(t+2j)!}.$$

Lemma 4 presents the integral form of $e^{-y}y^j$ expressed in terms of the Bessel functions. We present its proof in [23], which shines light on latter derivations.

**Lemma 4.** *[23] For any $j \in \mathbb{Z}^+$ and $y \ge 0$,*

$$e^{-y}y^j = \int_0^{\infty} e^{-\alpha}\alpha^j g_j(\alpha y)d\alpha.$$

*Proof.* By Fubini's theorem and the series expansion of $g_j$,

$$
\begin{aligned}
\int_0^{\infty} e^{-\alpha}\alpha^j g_j(\alpha y)d\alpha &= \int_0^{\infty} e^{-\alpha}\alpha^j \sum_{i=0}^{\infty} \frac{(-1)^i(\alpha y)^{i+j}}{i!(i+2j)!}d\alpha \\
&= \sum_{i=0}^{\infty} \frac{(-1)^i y^{i+j}}{i!(i+2j)!} \int_0^{\infty} e^{-\alpha}\alpha^{i+2j}d\alpha.
\end{aligned}
$$

Observe that the integral is actually $\Gamma(i + 2j + 1)$ and equals to $(i + 2j)!$,

$$\sum_{i=0}^{\infty} \frac{(-1)^i y^{i+j}}{i!(i+2j)!} \int_0^{\infty} e^{-\alpha} \alpha^{i+2j} d\alpha = \sum_{i=0}^{\infty} \frac{(-1)^i y^{i+j}}{i!(i+2j)!}(i+2j)!$$

$$= \sum_{i=0}^{\infty} \frac{(-1)^i y^{i+j}}{i!}$$

$$= e^{-y} y^j. \qquad \square$$

## E   Proof of Lemma 5

By the argument in Section 4, instead of estimating $U_\mu$, we will consider its Poissonized version $U_\mu^{\mathrm{P}}$ and approximate the value of $U_\mu^{\mathrm{P}}$ by a *near-unbiased* estimator of

$$\tilde{U}_\mu^{\mathrm{P}} \triangleq \sum_x \tilde{F}_\mu(a\lambda_x) \cdot e^{-\lambda_x},$$

where $\lambda_x \triangleq np_x$ and $\tilde{F}_\mu(y)$ is defined as $\sum_{j \leq \mu-1} \frac{1}{j!} f_j(y)$ for the sequence of real functions

$$f_0(y) = -\sum_{j=1}^{\infty} \frac{(-y)^j}{j!} \Pr(\mathrm{Poi}(r) \geq j) \quad \text{and} \quad f_j(y) = -\int_0^r e^{-\alpha} \alpha^j g_j(\alpha y) d\alpha, \ \forall j \geq 1.$$

The following lemma finds the series expansion of $\tilde{F}_\mu(y)$.

**Lemma 5.** *For any non-negative number $y$,*

$$\tilde{F}_\mu(y) = -\sum_{i=1}^{\infty} c_i \cdot \frac{y^i}{i!},$$

*where the coefficients are*

$$c_i \triangleq (-1)^i \Pr(\mathrm{Poi}(r) \geq i) + \sum_{j=1}^{(\mu-1)\wedge i} (-1)^{i-j} \binom{i}{j} \Pr(\mathrm{Poi}(r) \geq i + j + 1).$$

*Proof.* For any $j \geq 1$, the following equations hold.

$$f_j(y) \overset{(a)}{=} -\int_0^r e^{-\alpha} \alpha^j g_j(\alpha y) d\alpha$$

$$\overset{(b)}{=} -\int_0^r e^{-\alpha} \alpha^j \sum_{t=0}^{\infty} \frac{(-1)^j (\alpha y)^{t+j}}{t!(t+2j)!} d\alpha$$

$$\overset{(c)}{=} -\sum_{t=0}^{\infty} \frac{(-1)^j y^{t+j}}{t!(t+2j)!} \int_0^r e^{-\alpha} \alpha^{t+2j} d\alpha$$

$$\overset{(d)}{=} -\sum_{t=0}^{\infty} \frac{(-1)^t y^{t+j}}{t!} \Pr(\mathrm{Poi}(r) \geq t + 2j + 1),$$

where $(a)$ follows by the definition of $f_j$; $(b)$ follows by the series expansions of the Bessel functions; $(c)$ follows by Fubini's theorem; $(d)$ follows by the series expansions the incomplete Gamma functions. Therefore, we can write $\tilde{F}_\mu(y)$ as

$$\sum_{j \leq \mu-1} \frac{f_j(y)}{j!} = -\sum_{t=1}^{\infty} \frac{(-y)^t}{t!} \Pr(\mathrm{Poi}(r) \geq t) - \sum_{j=1}^{\mu-1} \frac{1}{j!} \sum_{t'=0}^{\infty} \frac{(-1)^{t'} y^{t'+j}}{t'!} \Pr(\mathrm{Poi}(r) \geq t' + 2j + 1)$$

$$= -\sum_{i=1}^{\infty} \left( (-1)^i \Pr(\mathrm{Poi}(r) \geq i) + \sum_{j=1}^{(\mu-1)\wedge i} \frac{(-1)^{i-j} i!}{j!(i-j)!} \Pr(\mathrm{Poi}(r) \geq i + j + 1) \right) \frac{y^i}{i!},$$

which completes the proof of the lemma. $\qquad \square$

## F  Lemma 6 and Bias Bound

Our construction closely relates to Bessel functions of the first kind. The following lemma provides useful bounds on their magnitude.

**Lemma 6.** *[1, 38] For any $s, y \geq 0$, we have $|J_s(y)| \leq 1$ and $|J_s(y)| \leq \frac{(y/2)^s}{s!}$.*

As a corollary, we have the following bounds for $g_j(y) = J_{2j}(2\sqrt{y})$, where $j \geq 0$.

**Corollary 1** (Extension of a result in [23]). *For $j \geq 0$ and $y \geq 0$, we have $|g_j(y)| \leq 1$, and if in addition, $j \geq 1$, then $|g_j(y)| \leq \frac{y}{j+1}$.*

*Proof.* The first inequality follows directly from Lemma 6. For the second inequality, by definition of $g_j$, we can therefore write

$$|g_j(y)| = |J_{2j}(2\sqrt{y})| \leq \frac{y^j}{(2j)!} \leq \frac{y^j}{j+1},$$

where the last step follows from $(j+1) \leq (2j)!$ for any $j \geq 0$. $\qquad\square$

Now, we are ready to establish the last bias bound in Section 4, which states that

$$\mathrm{Bias}_3 \triangleq \left| \tilde{U}_\mu^{\mathrm{P}} - \mathbb{E}[U_\mu^{\mathrm{P}}] \right| \leq \sum_x |\tilde{F}_\mu(a\lambda_x) - F_\mu(a\lambda_x)| \leq an(\mu+1) \cdot \mathrm{Pr}(\mathrm{Poi}(r) \leq \mu).$$

*Proof.* Recall that $\lambda_x$ abbreviates $np_x$. Consolidating the above bounds yields

$$\mathrm{Bias}_3 \overset{(a)}{\leq} \sum_x e^{-r} \left[ e^{-a\lambda_x} \int_0^{a\lambda_x} |g_0(sr)| e^s ds \right] + \sum_x \sum_{1 \leq j \leq \mu-1} \frac{1}{j!} \int_r^\infty e^{-\alpha} \alpha^j |g_j(\alpha a \lambda_x)| d\alpha$$

$$\overset{(b)}{\leq} \sum_x e^{-r} \left[ e^{-a\lambda_x} \int_0^{a\lambda_x} e^s ds \right] + \sum_x \sum_{1 \leq j \leq \mu-1} \frac{1}{j!} \int_r^\infty e^{-\alpha} \alpha^j \frac{\alpha a \lambda_x}{j+1} d\alpha$$

$$\overset{(c)}{=} e^{-r} \sum_x (1 - e^{-a\lambda_x}) + \sum_x a\lambda_x \sum_{1 \leq j \leq \mu-1} \frac{1}{(j+1)!} \int_r^\infty e^{-\alpha} \alpha^{j+1} d\alpha$$

$$\overset{(d)}{\leq} e^{-r} \sum_x a\lambda_x + e^{-r} \left( \sum_{j=1}^{\mu-1} \sum_{t=0}^{j+1} \frac{r^t}{t!} \right) \sum_x a\lambda_x$$

$$\overset{(e)}{\leq} e^{-r} an \left( 1 + \sum_{j=1}^{\mu-1} \sum_{t=0}^{j+1} \frac{r^t}{t!} \right)$$

$$\overset{(f)}{\leq} an(\mu+1) \cdot \mathrm{Pr}(\mathrm{Poi}(r) \leq \mu),$$

where $(a)$ follows by the triangle inequality; $(b)$ follows by Corollary 1; $(c)$ follows by simple algebra; $(d)$ follows by $1 - e^{-x} \leq x, \forall x \geq 0$ and the series expansion of the incomplete gamma function; $(e)$ follows by $\sum_x \lambda_x = n$; $(f)$ follows by adding non-negative terms and by Poisson probabilities. $\quad\square$

# G  Estimator Modification

In Section 4, we introduced a quantity $\tilde{U}_\mu^{\mathrm{P}}$ as our new approximation target, which takes the form of

$$\tilde{U}_\mu^{\mathrm{P}} = \sum_x \tilde{F}_\mu(a\lambda_x) \cdot e^{-\lambda_x} = -\sum_x \sum_{i=1}^{\infty} (c_i a^i) \cdot e^{-\lambda_x} \frac{\lambda_x^i}{i!} = -\sum_{i=1}^{\infty} (c_i a^i) \cdot \mathbb{E}[\Phi_i'],$$

where $\Phi_i'$ denotes the number of symbols appearing exactly $i$ times in the Poissonized sample, and the coefficients $c_i$'s are defined as

$$c_i \triangleq (-1)^i \Pr(\mathrm{Poi}(r) \geq i) + \sum_{j=1}^{(\mu-1)\wedge i} (-1)^{i-j} \binom{i}{j} \Pr(\mathrm{Poi}(r) \geq i+j+1), \ \forall i.$$

Under Poisson sampling, a simple unbiased estimator is $-\sum_{i=1}^{\infty}(c_i a^i)\cdot\Phi_i'$. It should also be clear that $s_i \approx c_i a^i$ for every $i$ as they are equal if we replace each smoothing probability $\Pr(\mathrm{Poi}(r)\geq i+j+1)$ by $\Pr(\mathrm{Poi}(r)\geq i+j)$ in the expression of $c_i$. Hence, given sample statistics $\Phi_i'$ for $i \geq 1$, we claimed that a *near-unbiased* estimator of $\tilde{U}_\mu^{\mathrm{P}}$ is

$$\hat{U}_\mu^{\mathrm{P}} = -\sum_{i=1}^{\infty} s_i \cdot \Phi_i'.$$

Below, we show that the absolute bias satisfies $|\mathbb{E}[\hat{U}_\mu^{\mathrm{P}}] - \tilde{U}_\mu^{\mathrm{P}}| \leq na\cdot\Pr(\mathrm{Poi}(r) \leq \mu)$ by leveraging the point-wise bounds on Bessel functions in the last section.

*Proof.* The difference between the two estimators is

$$|\mathbb{E}[\hat{U}_\mu^{\mathrm{P}}] - \tilde{U}_\mu^{\mathrm{P}}| = \sum_x d(a\lambda_x),$$

where $\lambda_x = np_x$ and

$$d(y) \triangleq -\sum_{j \leq \mu-1} \frac{1}{j!} \sum_{t=0}^{\infty} \frac{(-1)^t y^{t+j}}{t!} \Pr(\mathrm{Poi}(r) \geq t+2j).$$

First we show that $d(y)$ has a small magnitude. Specifically,

$$|d(y)| = \left| \sum_{j \leq \mu-1} \frac{1}{j!} \sum_{t=0}^{\infty} \frac{(-1)^t y^{t+j}}{t!} \Pr(\mathrm{Poi}(r) = t+2j) \right|$$

$$= \left| \sum_{j \leq \mu-1} \frac{1}{j!} \sum_{t=0}^{\infty} \frac{(-1)^t y^{t+j}}{t!} \left( e^{-r} \frac{r^{t+2j}}{(t+2j)!} \right) \right|$$

$$= \left| \sum_{j \leq \mu-1} e^{-r} \frac{r^j}{j!} \sum_{t=0}^{\infty} \frac{(-1)^t (yr)^{t+j}}{t!(t+2j)!} \right|$$

$$\stackrel{(a)}{=} \left| \sum_{j \leq \mu-1} e^{-r} \frac{r^j}{j!} \cdot g_j(yr) \right|$$

$$\stackrel{(b)}{\leq} \sum_{j \leq \mu-1} e^{-r} \frac{r^j}{j!} \cdot \frac{yr}{j+1}$$

$$= y \cdot \Pr(1 \leq \mathrm{Poi}(r) \leq \mu),$$

where the first three steps follow by algebraic manipulations; $(a)$ follows by the series expansion of $g_j$ (or equivalently the $2j$-th order Bessel function); $(b)$ follows by Corollary 1. Consequently, the absolute bias of our estimate satisfies

$$|\mathbb{E}[\hat{U}_\mu^{\mathrm{P}}] - \tilde{U}_\mu^{\mathrm{P}}| \leq \left| \sum_x d(a\lambda_x) \right| \leq \left( \sum_x a\lambda_x \right) \cdot \Pr(1 \leq \mathrm{Poi}(r) \leq \mu) \leq na \cdot \Pr(\mathrm{Poi}(r) \leq \mu). \quad \square$$

# H Lower Bounds

We establish a lower bound on the worst-case MSE of any estimator by connecting the task to the closely related task of support size estimation, which is ill-defined if we simply consider the collection of discrete distributions with bounded support size. Specifically, for an unknown distribution and any finite sample size $n$, there can be many symbols with tiny probabilities so that it is unlikely to observe any of them in a size-$n$ sample, yet they can have a huge impact on the distribution's support size.

Motivated by applications to database studies, the paper of [40] considered support size estimation for distributions whose positive probabilities are at least $1/k$, where $k$ is the known size of the alphabet (hence also serves as an upper bound on distributions' support sizes).

For any distribution $p \in \Delta$, denote by $S(p) = \sum_x \mathbb{1}_{p_x > 0}$ its support size, and by $p_{\min}^+$ its minimum nonzero probability. Paper [40] showed the following lower bound on the minimax MSE (together with a nearly matching upper bound).

**Lemma 9.** *Assume that $k \log k \gg n \gg \frac{k}{\log k}$ and $k \gg 1$,*

$$\min_{\hat{S}} \max_{p : p_{min}^+ \geq 1/k} \mathbb{E}_{X^n \sim p} \left( \hat{S}(X^n) - S(p) \right)^2 \geq \exp\left( -4\sqrt{\frac{n \log k}{k}} \right) k^2.$$

In the next two subsections, we will connect this estimation problem to ours, and establish Theorem 2.

## H.1 Connection between Two Problems

Recall that the quantity of interest is

$$U_\mu = \sum_x \mathbb{1}_{M_x \geq \mu} \cdot \mathbb{1}_{N_x = 0}.$$

For some integer $k$ to be determined later, assume that $p$ is a distribution satisfying $p_{\min}^+ \geq 1/k$. For the ease of exposition, define

$$U_\mu^p \triangleq \sum_x \mathbb{E}\left[ \mathbb{1}_{M_x \geq \mu} \right] \cdot \mathbb{1}_{N_x = 0} = \sum_x \Pr\left( \text{bin}(na, p_x) \geq \mu \right) \cdot \mathbb{1}_{N_x = 0},$$

Adding the number of observed symbols in $X^n$, we obtain

$$\tilde{U}_\mu^p \triangleq U_\mu(X^n, p) + \sum_x \mathbb{1}_{N_x > 0} = \sum_x \left( \Pr\left( \text{bin}(na, p_x) \geq \mu \right) \cdot \mathbb{1}_{N_x = 0} + \mathbb{1}_{N_x > 0} \right).$$

Below, we show that for any $\beta \geq 3$ and $k \leq na/(\beta \mu)$, and distribution $p$ satisfying $p_{\min}^+ \geq 1/k$,

$$\left| \tilde{U}_\mu^p - S(p) \right| \leq S(p) \exp\left( -0.3 \frac{na}{k} \right) \leq k \exp\left( -0.3 \frac{na}{k} \right).$$

*Proof.* The absolute difference between $\tilde{U}_\mu^p$ and $S(p)$ satisfies

$$S(p) - \tilde{U}_\mu^p \overset{(a)}{=} \sum_x \left( \mathbb{1}_{p_x > 0} \cdot \mathbb{1}_{N_x = 0} + \mathbb{1}_{N_x > 0} \right) - \sum_x \left( \Pr\left( \text{bin}(na, p_x) \geq \mu \right) \cdot \mathbb{1}_{N_x = 0} + \mathbb{1}_{N_x > 0} \right)$$

$$\overset{(b)}{=} \sum_x \mathbb{1}_{N_x = 0} \cdot \mathbb{1}_{p_x > 0} - \sum_x \Pr\left( \text{bin}(na, p_x) \geq \mu \right) \cdot \mathbb{1}_{N_x = 0} \cdot \mathbb{1}_{p_x > 0}$$

$$\overset{(c)}{=} \sum_x \left( 1 - \Pr\left( \text{bin}(na, p_x) \geq \mu \right) \right) \cdot \mathbb{1}_{N_x = 0} \cdot \mathbb{1}_{p_x > 0}$$

$$\overset{(d)}{\geq} 0,$$

where $(a)$ holds as $\mathbb{1}_{N_x > 0} = 0$ surely if $p_x = 0$; $(b)$ follows by term cancelation and the equivalence of $\Pr\left( \text{bin}(na, p_x) \geq \mu \right) = 0$ and $p_x = 0$, for any $1 \leq \mu \leq na$; $(c)$ follows by simply algebra; $(d)$ follows by $\Pr\left( \text{bin}(na, p_x) \geq \mu \right) \leq 1$.

On the other hand, for $k \leq na/(\beta\mu)$ and $\beta \geq 3$,

$$
\begin{aligned}
\tilde{U}_\mu^p &\overset{(a)}{=} \sum_x \left( \Pr\left(\mathrm{bin}(na, p_x) \geq \mu\right) \cdot \mathbb{1}_{N_x=0} + \mathbb{1}_{N_x>0} \right) \\
&\overset{(b)}{=} \sum_{x:p_x \geq 1/k} \left( (1 - \Pr\left(\mathrm{bin}(na, p_x) < \mu\right)) \cdot \mathbb{1}_{N_x=0} + \mathbb{1}_{N_x>0} \right) \\
&\overset{(c)}{\geq} \sum_{x:p_x \geq 1/k} \left( 1 - \Pr\left(\mathrm{bin}(na, 1/k) < \mu\right) \cdot \mathbb{1}_{N_x=0} \right) \\
&\overset{(d)}{\geq} \sum_{x:p_x \geq 1/k} \left( 1 - \Pr\left(\mathrm{bin}(na, 1/k) < \frac{na}{3k}\right) \cdot \mathbb{1}_{N_x=0} \right) \\
&\overset{(e)}{\geq} \sum_{x:p_x \geq 1/k} \left( 1 - \exp\left(-0.3\frac{na}{k}\right) \cdot \mathbb{1}_{N_x=0} \right) \\
&\overset{(f)}{\geq} S(p)\left( 1 - \exp\left(-0.3\frac{na}{k}\right) \right),
\end{aligned}
$$

where $(a)$ follows by the definition of $\tilde{U}_\mu^p$; $(b)$ follows by the equivalence of $p_x = 0$, $\mathbb{1}_{N_x>0} = 0$ and $\Pr\left(\mathrm{bin}(na, p_x) \geq \mu\right) = 0$; $(c)$ follows by $\mathbb{1}_{N_x=0} + \mathbb{1}_{N_x>0} = 1$; $(d)$ follows by the two conditions $k \leq na/(\beta\mu)$ and $\beta \geq 3$; $(e)$ follows by standard Chernoff bound for binomial random variables; $(f)$ holds as the summation is over $x$ satisfying $p_x \geq 1/k$.

Consequently, we have shown that

$$
\left| \tilde{U}_\mu^p - S(p) \right| \leq S(p) \exp\left(-0.3\frac{na}{k}\right) \leq k \exp\left(-0.3\frac{na}{k}\right). \qquad \square
$$

## H.2 Proof of Theorem 2

**Theorem 2.** *There exist absolute constants $c'$ and $c_0'$ such that for any $a \geq (c' \log n)/\mu$, and any $n$-sample estimator $\hat{U}$,*

$$
\mathcal{E}_{n,a}^\mu(\hat{U}) \gtrsim \frac{1}{n^{c_0'/a}}.
$$

The theorem lowerly bounds the worst-case normalized MSE of the best estimator.

*Proof.* For any estimator $\hat{U}$ and distribution $p$, the MSE of estimating $U_\mu$ admits

$$
\begin{aligned}
\mathrm{MSE} &\overset{(a)}{=} \mathbb{E}\left(\hat{U} - U_\mu\right)^2 \\
&\overset{(b)}{\geq} \underset{X^n \sim p}{\mathbb{E}} \left(\hat{U} - U_\mu^p\right)^2 \\
&\overset{(c)}{\geq} \frac{1}{2} \underset{X^n \sim p}{\mathbb{E}} \left(\hat{U} - U_\mu^p + \tilde{U}_\mu^p - S(p)\right)^2 - \underset{X^n \sim p}{\mathbb{E}} \left(\tilde{U}_\mu^p - S(p)\right)^2 \\
&\overset{(d)}{\geq} \frac{1}{2} \underset{X^n \sim p}{\mathbb{E}} \left(\hat{U} + \sum_x \mathbb{1}_{N_x>0} - S(p)\right)^2 - \underset{X^n \sim p}{\mathbb{E}} \left(\tilde{U}_\mu^p - S(p)\right)^2 \\
&\overset{(e)}{\geq} \frac{1}{2} \underset{X^n \sim p}{\mathbb{E}} \left(\hat{U} + \sum_x \mathbb{1}_{N_x>0} - S(p)\right)^2 - \exp\left(-0.6\frac{na}{k}\right) k^2,
\end{aligned}
$$

where $(a)$ follows by definition; $(b)$ follows by the Jensen's inequality; $(c)$ follows by the linearity of expectation and inequality $a^2 \geq \frac{1}{2}(b+a)^2 - b^2$; $(d)$ follows by $\tilde{U}_\mu^p - U_\mu^p = \sum_x \mathbb{1}_{N_x>0}$; $(e)$ follows by the main result of Appendix H.1 and our choice of parameters.

Consider $a\mu = c' \log n$ for some $c' \gg 1$. For $a, \mu \ll \log n$, if we choose $k = \frac{na}{\beta\mu}$, then $\frac{k}{\log k} \ll n \ll k \log k$. Therefore, by Lemma 9 on support size estimation, the minimax MSE of learning $U_\mu$, which we refer to as the MMSE, satisfies

$$
\begin{aligned}
\text{MMSE} &\stackrel{(a)}{=} \min_{\hat{U}} \max_{p \in \Delta} \mathbb{E} \left( \hat{U} - U_\mu \right)^2 \\
&\stackrel{(b)}{\geq} \min_{\hat{U}} \max_{p:\, p_{\min}^+ \geq 1/k} \mathbb{E} \left( \hat{U} - U_\mu \right)^2 \\
&\stackrel{(c)}{\geq} \frac{1}{2} \min_{\hat{U}} \max_{p:\, p_{\min}^+ \geq 1/k} \mathbb{E}_{X^n \sim p} \left( \hat{U} + \sum_x \mathbb{1}_{N_x > 0} - S(p) \right)^2 - \exp\left( -0.6 \frac{na}{k} \right) k^2 \\
&\stackrel{(d)}{\geq} \frac{1}{2} \exp\left( -4 \sqrt{\frac{n \log k}{k}} \right) k^2 - \exp\left( -0.6 \frac{na}{k} \right) k^2,
\end{aligned}
$$

where $(a)$ follows by the definition; $(b)$ follows by maximizing over a subset set of the original; $(c)$ follows by the inequality that we established above; and $(d)$ follows by the lower bound in Lemma 9 and viewing $\hat{U} + \sum_x \mathbb{1}_{N_x > 0}$ as an $n$-sample support size estimator.

Choose $\beta = (10 \log 4)/\mu + 3$ so that we have both $\beta \geq 3$ for any $\mu \geq 1$ and $0.1 \geq (\log 4)/(\beta\mu)$. To ensure non-triviality, we require

$$
\exp\left( -4\sqrt{\frac{n \log k}{k}} \right) k^2 \geq 4 \exp\left( -0.6 \frac{na}{k} \right) k^2 \impliedby -4\sqrt{\frac{n \log k}{k}} \geq \log 4 - 0.6 \frac{na}{k}
$$

$$
\iff 4\sqrt{nk \log k} \leq 0.6na - na\frac{\log 4}{\beta\mu}
$$

$$
\impliedby 4\sqrt{nk \log k} \leq 0.5na
$$

$$
\impliedby \log n \ll a\mu,
$$

where the last step follows by $1 \leq a, \mu \ll n$. By definition, we have $k = M_\mu/\beta$. Therefore,

$$
\min_{\hat{U}} \max_{p \in \Delta} \mathbb{E} \left( \hat{U} - U_\mu \right)^2 \geq \exp\left( -0.6\mu \right) k^2 = \left( \frac{1}{\beta} M_\mu e^{-0.3\mu} \right)^2.
$$

Dividing both sides by $M_\mu^2$ yields

$$
\begin{aligned}
\min_{\hat{U}} \max_{p \in \Delta} \mathbb{E}_{X^n \sim p} \left( \frac{\hat{U} - U_\mu}{M_\mu} \right)^2 &\geq \left( \frac{1}{\beta} e^{-0.3\mu} \right)^2 \\
&= \left( \frac{1}{\beta} \exp\left( -0.3 \frac{c' \log n}{a} \right) \right)^2 \\
&\gtrsim \frac{1}{n^{c_0'/a}},
\end{aligned}
$$

where $c_0' \triangleq 0.6c'$ is a positive absolute constant. This completes the proof of Theorem 2. $\qquad \square$

# I  Experiments on LOTR (Lord of the Rings) Datasets

The task is *vocabulary size estimation*, one that we mentioned at the beginning of Section 1.1. Replacing butterflies by words, *vocabulary size estimation* [9, 11, 19, 35] aims to determine how many words a writer, say William Shakespeare, knew based on his written works. An intuitive and widely used approach is to simply add up the number of observed (distinct) words and some estimate of $U_1$. With the same motivation, we may also want to know how many words fall into a writer's *common vocabulary* (excluding those that appear only once or twice), which calls for estimating $U_\mu$.

Specifically, the book we used in the experiments is *Lord of the Rings*, written by J. R. R. Tolkien. We found a specific version of this book consisting of 573,479 words, of which 16,335 are distinct. To split the data into two parts, we randomly selected $1/(a+1)$ fraction of the words, and applied our estimator to estimate the number of *new* words that appear at least $\mu$ times in the remaining text. Since $n$ is relatively small, we considered $\mu \le 4$. We simply choose $r = \frac{\log n}{a}$ in the experiments.

Figure 2: We presents the results for $\mu \in \{1, 2, 3, 4\}$ and the sample amplification ratio $a \le 1$. The dashed (black) curves represent the truth while the solid (red) curves represent our estimate (the mean). The shaded areas illustrate the standard deviation from the mean. For $\mu = 3$ and $\mu = 4$, since the standard deviations become so large, we repeat them (the bottom two plots) by restricting the y-axes to be smaller, to better visualize the results.

As shown in Figure 2, the proposed estimator provides accurate estimates for $a \leq 1$ and $\mu = 1$. For $\mu = 2$, we can see that the standard deviation becomes much larger when $a$ approaches 1. For $\mu = 3$ and $\mu = 4$, the estimates are still fairly accurate when $a \leq 0.7$. However, for larger $a$ values, the standard deviations become so large that the estimates may not be very reliable.

Figure 3 presents additional plots for $\mu = 1$ and $\mu = 2$, and for $a$ up to 4. For $\mu = 1$, the the estimates are still fairly reliable when $a \leq 2$.

Figure 3: We presents the results for $\mu \in \{1, 2\}$ and the sample amplification ratio $a$ up to 4. The dashed (black) curves represent the truth while the solid (red) curves represent our estimate (the mean). The shaded areas illustrate the standard deviation from the mean.

# J  Derivation of the Generalized G-T Estimator for $\mu \geq 1$

In this section, we derive a new estimator which generalizes the G-T estimator for $\mu \geq 1$:

$$\hat{U}_{\text{GGT}} \triangleq (-1)^{\mu} \sum_{i=\mu}^{n} (-a)^i \binom{i-1}{\mu-1} \Phi_i$$

We obtain this estimator by letting $r \to \infty$ in the proposed estimator using Poisson smoothing:

$$\hat{U}_{\mu} \triangleq \hat{U}_{\mu}(X^n, a) \triangleq \sum_{i=1}^{n} s_i \cdot \Phi_i$$

where

$$s_i \triangleq -\sum_{j=0}^{(\mu-1)\wedge i} (-a)^i (-1)^j \binom{i}{j} \Pr(\text{Poi}(r) \geq i+j)$$

When $r \to \infty$, we have $\Pr(\text{Poi}(r) \geq i+j) = 1$, and the coefficients become

$$s_i \triangleq -(-a)^i \sum_{j=0}^{(\mu-1)\wedge i} (-1)^j \binom{i}{j} = -(-a)^i h_{\mu,i} \quad \text{where } h_{\mu,i} = \sum_{j=0}^{(\mu-1)\wedge i} (-1)^j \binom{i}{j}$$

The estimator can then be re-written as

$$\hat{U}_{\mu} \triangleq -\sum_{i=1}^{n} (-a)^i h_{\mu,i} \cdot \Phi_i$$

We can prove that, when $i \leq \mu - 1$, we always have $h_{\mu,i} = 0$, because

$$h_{\mu,i} = \sum_{j=0}^{i} (-1)^j \binom{i}{j} = \sum_{j=0}^{i} (-1)^j (1)^{i-j} \binom{i}{j} = (1-1)^i = 0$$

Therefore, we can write

$$h_{\mu,i} = \begin{cases} 0 & i \leq \mu - 1 \\ \sum_{j=0}^{\mu-1} (-1)^j \binom{i}{j} & i > \mu - 1 \end{cases}$$

When $\mu = 3$ and $i > 2$, we have

$$h_{3,i} = 1 - i + \frac{i(i-1)}{2} = \frac{(i-1)(i-2)}{2} = \binom{i-1}{2}$$

When $\mu = 4$ and $i > 3$, we have

$$h_{4,i} = -(i-1) + \frac{i(i-1)}{2} - \frac{i(i-1)(i-2)}{6} = -\frac{(i-1)(i-2)(i-3)}{3 \cdot 2} = -\binom{i-1}{3}$$

Assume that, for $i > \mu - 1$,

$$h_{\mu,i} = (-1)^{\mu-1} \binom{i-1}{\mu-1}$$

Then

$$\begin{aligned} h_{\mu+1,i} &= (-1)^{\mu-1} \binom{i-1}{\mu-1} + (-1)^{\mu} \binom{i}{\mu} = (-1)^{\mu} \left\{ \binom{i}{\mu} - \binom{i-1}{\mu-1} \right\} \\ &= (-1)^{\mu} \left\{ \frac{i!}{\mu!(i-\mu)!} - \frac{(i-1)!}{(\mu-1)!(i-\mu)!} \right\} \\ &= (-1)^{\mu} \frac{(i-1)!}{\mu!(i-\mu-1)!} \left\{ \frac{i}{i-\mu} - \frac{\mu}{i-\mu} \right\} = (-1)^{\mu} \binom{i-1}{\mu} \end{aligned}$$

This proves, by induction, that $h_{\mu,i} = (-1)^{\mu-1} \binom{i-1}{\mu-1}$, and hence we can write the estimator as

$$\hat{U}_{\text{GGT}} = (-1)^{\mu} \sum_{i=\mu}^{n} (-a)^i \binom{i-1}{\mu-1} \Phi_i$$