[Reviews · NeurIPS 2020]

Review 1

Summary and Contributions: This paper considers the following problem. Given n samples from an unknown distribution, estimate the number of symbols that will appear at least mu times in a new sample of size a*n. In particular, they consider the largest ‘a’ (As a function of mu) for which it is possible to estimate it. They characterize this up to a constant factor. The paper considers a clean problem, is very well written, well motivated, easy to understand, and is of general interest to a broader audience. The problem could also find potential applications in other fields. The problem should be a little better motivated. In particular, it is a natural extension of the unseen species problem which is the special case when mu=1, namely how many unseen symbols will we see in the future sample? I like the example of number of users that will use the app at least a few times. The technical parts of the paper seem to be straight-forward extensions of the prior results of HaoLi, HaoOrlitsky, and OrlitskySureshWu. Please explain what the additional technical insights are. This is perhaps the reason for my current score. I am unable to appreciate the technical contributions, and as of now, the plus point of the paper seems to be a new problem extending to larger mu, but the novelty in the technical contributions seems unclear to me. I would appreciate the authors providing a detailed reason for the novelty in the proposed approach.

Strengths: See above.

Weaknesses: See above.

Correctness: Yes.

Clarity: Yes, very well written.

Relation to Prior Work: Yes.

Reproducibility: Yes

Additional Feedback:


Review 2

Summary and Contributions: Consider the classical statistical problem of estimating the number of unseen species. Specifically, given n samples from an unknown discrete distribution, we wish to predict how many unseen symbols will be observed should we collect m additional independent samples. A 2016 paper of Orlitsky, Suresh, and Wu established that the smoothed Good-Toulmin estimator is minimax optimal (up to constants). This paper generalizes these results considerably along the following axis: given n samples, how many unseen symbols will be observed \mu times in another independent m samples. The authors propose a generalization of the smoothed Good-Toulmin estimator, establish its minimax optimality (up to constants), and demonstrate good agreement between their theoretical prediction and empirical studies on synthetic and real datasets.

Strengths: This is an excellent theoretical paper, and in my view, checks all of the boxes for a good theory paper. Specifically, it 1) poses a novel and interesting mathematical question (estimating the number of unseen symbols that will be observed \mu times), motivated by real-world applications, 2) places this question in the context of prior work, 3) proposes a solution, 4) establishes matching upper and lower bounds with clear, concise, and intuitive proofs, and 5) validates its theoretical predictions with numerical experiments on synthetic and real datasets. In particular, the question of estimating the number of new symbols that will be observed \mu times in a future sample is quite interesting, and I was surprised to learn that this had not been carefully studied in the literature. This paper closes this gap.

Weaknesses: Edit after author feedback: I appreciate that the authors will add experiments for large \mu values, which I believe addresses the comment below. From my reading, the main weakness is that the focus, especially with regards to numerical experiments, is on small values of \mu (the number of occurrences). Given that this problem has not been carefully studied in the past, I would think that the regime where \mu is relatively large would be of interest. I realize that this might be a challenging regime in terms of computation, but it would be nice to have some reflection on how the small and large \mu regimes might differ, at least as predicted by the theory.

Correctness: To the best of my understanding, the proofs are correct and the numerical experiments are sound.

Clarity: The paper is very well-written, with the appropriate level of detail and intuition for the 9-page version (shuffling details off to the supplement).

Relation to Prior Work: The paper is very carefully and clearly placed within the context of prior work.

Reproducibility: Yes

Additional Feedback:


Review 3

Summary and Contributions: This work generalizes naturally the classical problem of estimating the number of unseen species. From a sample of length n, the task is to estimate the number of new symbols that would appear mu times if we were to have access to another sample of size m = a*n. The authors fully characterize the amplification ratio a for which this number of unseen symbols can be inferred successfully. The further illustrate their claim with both synthetic and real world data. === AFTER REBUTTAL === The reviewer thanks the authors for their detailed reply, and keeps the score unchanged (7: Accept).

Strengths: * The problem is well-motivated, with clear real world applications. * The generalization proposed by the authors is relevant. * The techniques are non-trivial and can possibly find application elsewhere. * The result is crisp, characterizing the amplification ratio that allows for estimation. * An empirical evaluation based on synthetic data illustrates the claims. Moreover, in the supplementary material, the authors also successfully apply their method to real world data, for the task of vocabulary task estimation.

Weaknesses: * No particularly strong weakness. * The reviewer would have liked to have a more precise idea of the constants c0 and c0' in Theorem 1 and 2.

Correctness: * In terms of correctness, although the reviewer did not carefully check the proofs in the supplementary material, he also has no reason to doubt them. * The results recover the known ones in the simpler setting studied in [23].

Clarity: * The paper is neatly written, the reviewer enjoyed reviewing this work. * The proof strategy is well explained.

Relation to Prior Work: The considered estimator is well introduced, as is its link to previous work.

Reproducibility: Yes

Additional Feedback: * For Theorem 1 and Theorem 2, could you please add explicit bounds for the universal constants c, c', c0 and c0' ? * Typos: L50: "Susequently" -> "Subsequently" L81: "smoothd" -> "smoothed" L140: "inquality" -> "inequality" L144: "not sensitive" -> "not too sensitive" L148: "significant" -> "significantly" L150: "we first finds" -> "we first find" L355: "inquality" -> "inequality"


Review 4

Summary and Contributions: See report below.

Strengths: See report below.

Weaknesses: See report below.

Correctness: Yes to the best of my knowledge.

Clarity: Yes.

Relation to Prior Work: Yes.

Reproducibility: Yes

Additional Feedback: This paper studies a variant of Fisher et al's unseen species problem, namely, predicting the number of new symbols that appears at least \mu times in the future (unobserved) sample of size a \times n on the basis of the existing sample of size n. This extends the results of Orlitsky et al. [22] focusing on \mu=1, the original setting in Fisher et al. The main findings are - Theorem 1: an estimator is constructed using the smoothing technique from [22] that achieves a normalized prediction error of n^{-\Omega(1/a)} provided a = O(log n/mu) - Theorem 2: a minimax lower bound n^{-O(1/a)} is shown, provided a = \Omega(log n/mu). Both the construction and the analysis follow closely those in [22]. Namely, the upper bound is obtained by following the recipe of smoothed estimator (by modifying the unbiased estimator) and the analysis uses Poisson sampling and relies on Bessel function to control the bias from cancellation; the lower bound is obtained by a reduction to the support size estimation problem. Nevertheless, the results are interesting with its own applications (as the authors nicely argued in Sec 1.1) and the extensions are technically non-trivial. My main comments are as follows: - I did not understand the meaning of the word "reproducibility" in the title and I wonder whether it is an appropriate choice of words. In the scientific communities, especially this day and age, "reproducibility" typically means whether a given piece of research can be reproduced by others (see https://academic.oup.com/biostatistics/article/11/3/385/257703, https://www.nature.com/news/1-500-scientists-lift-the-lid-on-reproducibility-1.19970, https://www.nature.com/collections/prbfkwmwvz/). Since the current title = title of [22] + "with reproducibility", it is unclear what it really means. Given the novelty here is to consider mu>1 as opposed to mu=1, perhaps replacing "reproducibility" by "multiplicity" makes more sense. - I did not understood why the authors claim ``We completely resolve this problem by determining the limit of estimation to be a ≈ (logn)/µ''. When a = (logn)/µ, the lower bound of Thm 2 reads exp(- const* mu), which can still goes to zero if mu goes to infinity. It does not look like you are assuming mu being a constant throughout and I believe Thm 1 applies to arbitrary mu. - One thing I find curious is the following: the lower bound in [22] is based on reduction to the support size estimation problem. This is intuitively understandable because if one can predict the number of unseen symbols (mu=1) well then one can estimate the size of the entire support well. I did not get how this logic works for mu>1. This is not explained in the paper. Other comments: - The following papers are relevant which authors should consider citing: https://arxiv.org/abs/1902.05616 This paper determined the optimal exponent C(a) for the unseens species problem (\mu=1) in n^{-C(a)} for constant a. This improves (and seems out of the reach of) smoothing-based estimators. - Lemma 6. I did not understand what is the role of j \geq 1. - Typos and grammatical issues: + p2, The original introduction paper [11] for \hat{U}_GT_ -> The paper [11] that originally introduced \hat{U}_GT + p6, ``It is worth mentioning that the induced results in [ 16 ]'' -- what does "induced results" mean? + p6 last display, E[] should be P[] ? + In all figures on p8, horitonal axis t should be a

[Author Response · NeurIPS 2020]

We thank the reviewers for their thoughtful and helpful feedback. We carefully went through the paper again in the past few days and fixed all the typos and grammatical issues mentioned in the reviews. Below we will address the comments regarding our results and techniques.

**Reviewer 1**    Thank you again for your insightful and positive comments.

*The problem should be a little better motivated.* Agree. We motivated the problem in multiple ways – as a novel problem in functional estimation (line 14 to 21), as a generalization of the well-known Fisher's unseen species problem (line 22 to 57), and as an interesting task with numerous practical applications (line 65 to 74). Nevertheless, we want to motivate the problem even better by adding examples of genetic research and scores of basketball players to the introduction. We will also add the respective real-data experiments to the appendices.

*Technical novelty* Yes, our estimator's construction makes use of some prior results in [15], [16], and [22]. The combination might seem relatively simple in retrospect, but requires someone first to have the insight and realize the potential of the methods. In particular, one needs to properly manipulate each estimator component and provide tight analyses for both the upper and lower bounds. Besides, we made an effort to simplify the estimator (Appendix G) and multiple proofs (e.g., Appendices F and H) for technical beauty.

**Reviewer 2**    Thank you for the thoughtful and encouraging feedback, and for pointing out the importance of under-standing how a large $\mu$ value will affect the practical performance of the estimator. The theoretical analysis already showed that the variance will become larger as $\mu$ grows. We will add additional experiments for the large-$\mu$ regime and comment on the results accordingly.

**Reviewer 3**    Thank you for the thorough and helpful comments. We really appreciate your effort in helping us correct the typos and improve the clarity of the theorems.

*Bounds on universal constants* Currently, the lower-bound constant $c_0'$ is $0.4$, and the upper-bound constant $c_0$ is $3.0$. These might not be the best constants we can get because they were chosen to simplify the proofs. We will make all the four constants ($c, c_0, c'$, and $c_0'$) explicit in the draft and also work on optimizing their values.

**Reviewer 4**    Thank you for your constructive and valuable insights.

*Replacing the word "reproducibility"* Yes, we agree with your opinion on the choice of words. Thanks for also providing the references on the word "reproducibility" and its usage in the scientific communities. We have modified our paper accordingly and replaced "reproducibility" by "multiplicity."

*Claim on "resolv(ing) this problem"* The claim simply refers to the fact that the lower bound matches the upper bound, up to constants in the exponents. The theorems require $\mu = \mathcal{O}(\log n)$, which translates to something like $a \geq 1$ for $a \approx \log n / \mu$. Otherwise, the problem becomes relatively simple as we extrapolate no more than what has been observed. The condition of $a \geq 1$ was also required in our primary references [16, 22] and termed as "the interesting case" in the arXiv paper you mentioned. We appreciate the sharp observation and have updated the draft accordingly.

*Lower bound proof and support size estimation* Here is an over-simplified yet intuitive explanation. The paper [32] on support size estimation constructed nontrivial lower bounds for $m/\log m \ll k \ll m \log m$, where $m$ is the sample size and $1/k$ is a lower bound on the minimum positive probability of the underlying distribution. Hence, for $\mu \ll \log m$, one can adjust $k$ to be something like $m/\mu$. Then, we can leverage the results in [32] and work in the regime where every symbol in the extended sample (unseen) will appear at least $\mu$ times in expectation.

*Citing the paper: https://arxiv.org/abs/1902.05616.* Sure, we will cite the paper and comment on it appropriately.

*The role of "$j \geq 1$" in Lemma 6.* A typo, $j$ should be $s$ in the lemma. In fact, we removed this as it is unnecessary.

[Meta-Review · NeurIPS 2020]

This paper studies a variant of Fisher et al's unseen species problem, namely, predicting the number of new symbols that appears at least \mu times in the future (unobserved) sample of size a \times n on the basis of the existing sample of size n. A minimax optimal strategy (up to a constant factor) with corresponding upper and lower bounds are presented for this problem.